# COVID-19 prevention is shaped by polysocial risk: A cross-sectional study of vaccination and testing disparities in underserved populations

David R. Brown[iD]1*, Derek D. Cyr2, Lisa Wruck2, Troy A. Stefano1, Nader Mehri3, Zoran Bursac4, Richard Munoz4, Marianna K. Baum[iD]4, Eileen Fluney5, Prasad Bhoite[iD]1, Nana Aisha Garba1, Frederick W. Anderson[iD]1, Haley R. Fonseca[iD]4, Sara Assaf6, Krista M. Perreira7*

**1** Herbert Wertheim College of Medicine, Florida International University, Miami, Florida, United States of America, **2** Duke Clinical Research Institute, Duke University School of Medicine, Durham, North Carolina, United States of America, **3** Center for Health Equity Research, University of North Carolina School of Medicine, Chapel Hill, North Carolina, United States of America, **4** Robert Stempel School of Public Health and Social Work, Florida International University, Miami, Florida, United States of America, **5** Paradise Christian School, Hialeah, Florida, United States of America, **6** University of New Mexico, Albuquerque, New Mexico, United States of America, **7** Department of Social Medicine, University of North Carolina School of Medicine, Chapel Hill, North Carolina, United States of America

* drbrown@fiu.edu (DRB); krista_perreira@med.unc.edu (KMP)

## Abstract

Understanding disparities in COVID-19 preventive efforts among underserved populations requires a holistic approach that considers multiple social determinants of health (SDOH). While disparities in individual COVID-19 risk factors are well-documented, the cumulative impact of these factors on vaccine uptake and testing remains insufficiently quantified. This study applies a polysocial risk framework to assess the combined influence of geo-demographic, economic, and health-related factors on COVID-19 vaccination and testing. Using cross-sectional data from 9,758 participants enrolled in the NIH Rapid Acceleration of Diagnostics – Underserved Populations (RADx-UP) program (February 2020–April 2023), we analyzed associations between polysocial risk and preventive behaviors using multivariable generalized estimating equations (GEE). Overall, 72.5% of participants reported COVID-19 vaccination, and 82.1% reported testing. However, disparities were evident across polysocial risk profiles. Individuals experiencing intersecting geo-demographic (Non-Hispanic Black, age 45, Southern residence), economic (low education, unemployment, financial hardship), and health-related risk factors (substance use, low CVD risk, no flu vaccination) were 43−48 percentage points less likely to be vaccinated compared to groups with higher adoption (p < 0.001). Testing disparities were narrower but remained significant, with differences ranging from 2 to 27 percentage points depending on the specific polysocial risk profiles. The findings underscore the utility of polysocial risk modeling as a predictive tool for identifying populations at highest risk of disengagement from preventive care, informing targeted precision

**Data availability statement:** This study makes use of data collected by individual RADx-UP projects and shared with the CDCC under the RADx-UP data sharing policy which provides multi-level protections for confidentiality of participants. De-identified data underlying the results presented in this analysis are available by request from the NIH Rapid Acceleration of Diagnostics Data Hub (RADx Data Hub) https://radxdatahub.nih.gov/.

**Funding:** Research reported in this RADx-UP publication was supported by the National Institutes of Health under Award Numbers U01DA040381, U01MD017423, and U24MD016258. This work was supported by Azure sponsorship credits granted by Microsoft's AI for Good Research Lab. The content is solely the responsibility of the authors and does not necessarily represent the official views of the National Institutes of Health. The funders had no role in study design, data collection and analysis, decision to publish, or preparation of the manuscript.

**Competing interests:** The authors have declared that no competing interests exist.

public health interventions. Beyond COVID-19, this approach has broader applicability for understanding disparities in chronic disease prevention, cancer screening, maternal and child health, and health-related social needs (HRSN) interventions. Integrating polysocial risk assessments into clinical and public health settings can enhance data-driven strategies to improve population health outcomes.

## Introduction

Underserved populations experience persistent health disparities [1] driven by interconnected social determinants of health (SDOH)—structural and contextual factors such as economic stability, education, healthcare access, and neighborhood environment [2,3]. These disparities were particularly evident during the COVID-19 pandemic, where variations in healthcare access, economic constraints, and logistical barriers led to differences in vaccination, testing, and mortality rates [4–10]. While traditional models like Behavioral Model of Health Service Use [11,12] and the Health Beliefs Model (HBM) [13] have identified key demographic, socioeconomic, and health-related influences on preventive behavior adoption [14–21], they do not fully account for the complexity of multiple intersecting social determinants.

The polysocial risk framework, analogous to polygenic risk scores in genetics [22], provides a structured method for evaluating how social determinants interact to shape health outcomes and influence health behaviors collectively [23–28]. This approach aligns with Bourdieu's theory of capital, which conceptualizes economic, social, and cultural resources as key determinants of individual opportunities and health prospects [29,30]. As public health efforts increasingly rely on data-driven approaches, precision public health (PPH) has emerged as a framework for tailoring interventions at a population level [31–34]. However, PPH faces a significant challenge in quantifying the combined effects of multiple social determinants. The polysocial risk framework addresses this gap by providing a systematic method for assessing how intersecting social factors influence health behaviors. This approach is particularly relevant as healthcare systems expand screening for health-related social needs (HRSN) [35] and implement broader strategies to reduce disparities at scale [36–39].

The NIH-funded Rapid Acceleration of Diagnostics – Underserved Populations (RADx-UP) program provides an ideal opportunity to study these relationships in real-world community settings. This national initiative supports community-engaged research addressing gaps in COVID-19 testing and vaccination, enabling examination of how social risk factors influence prevention behaviors across diverse populations and contexts [40–42]. Using RADx-UP data, this study applies the polysocial risk framework to assess disparities in COVID-19 vaccination and testing. Our objectives are to: (1) quantify the prevalence of COVID-19 prevention behaviors, (2) evaluate how various social determinants influence these behaviors, (3) assess the contributions of different social factor combinations, and (4) identify subgroups

experiencing multiple barriers to prevention. By examining how multiple social determinants shape COVID-19 prevention behaviors, this study moves beyond individual risk factors to identify high-risk subgroups for targeted interventions. In doing so, it addresses a critical gap in understanding how multiple social factors collectively influence prevention efforts.

## Methods

### Study design and data source

This cross-sectional study uses data collected by individual RADx-UP projects between February 3, 2020 and April 21, 2023 and integrated by the Coordination and Data Collection Center (CDCC) [41]. The CDCC maintains a standardized data repository with common data elements (CDEs) from RADx-UP projects, allowing for cross-project analysis of factors influencing COVID-19 outcomes in underserved communities [40,43]. The contributing projects varied in geographic location, study design, and population characteristics, with further details available in S1 Table.

This study used data from the RADx-UP Core Analytic Datasets version 1.6 accessed through the CDCC on April 30, 2023 [40,43]. The first available observation for each participant was used for analysis. The components of the CDEs used were: Sociodemographic, Location, Covid Test, Vaccine Acceptance, Testing, Housing/Employment/Insurance, Alcohol and Tobacco, Work PPE and Distancing, Medical History, and Health Status. In this study, we conducted an integrated analysis of geo-demographic context, economic characteristics, and health-related characteristics. This approach aims to provide a more thorough understanding of the barriers and facilitators to prevention engagement.

All participants provided written informed consent for sharing data with the CDCC for analysis. IRB approval was obtained at the institutional level for each individual RADx-UP project and at the Duke IRB (PRo00106873) for the RADx-UP CDCC.

### Participants

Participants were adults aged 18 years and older who underwent COVID-19 testing and were enrolled in a RADx-UP project. A series of project-level and participant-level exclusions were applied to derive the final study population, as detailed in Fig 1 (Consort Flow Diagram) and S1 Appendix. Exclusion Criteria.

### Measures

The primary outcome measures in this study were COVID-19 vaccination status and COVID-19 testing engagement. Vaccination status was determined based on participant responses to the question, "Have you received a COVID-19 vaccine?" COVID-19 testing engagement was assessed through participants' history of testing ("Have you ever been tested for COVID-19?") prior to their enrollment in the RADx-UP study. Prior test positivity rates were also examined among participants who had previously engaged in testing.

In this study, 'health risk groups' describe individuals with characteristics affecting their likelihood of engaging in health-promoting activities, such as COVID-19 vaccination and testing. Higher health risk groups are those with characteristics associated with lower uptake of preventive measures, while lower health risk groups have characteristics associated with higher uptake.

Independent variables were selected to represent key social factors influencing prevention behaviors, spanning three domains:

(1) **Geo-demographic context**, including age, sex assigned at birth, race/ethnicity, and U.S. geographic region;

(2) **Economic characteristics**, such as education, household income, employment status, household description, economic challenges; and

(3) **Health-related characteristics**, encompassing insurance status, health status, disability, any illicit drug use, heavy alcohol use, any mental health problem, cardiometabolic risk, other chronic conditions, access to COVID-19 testing, prior flu vaccination history.

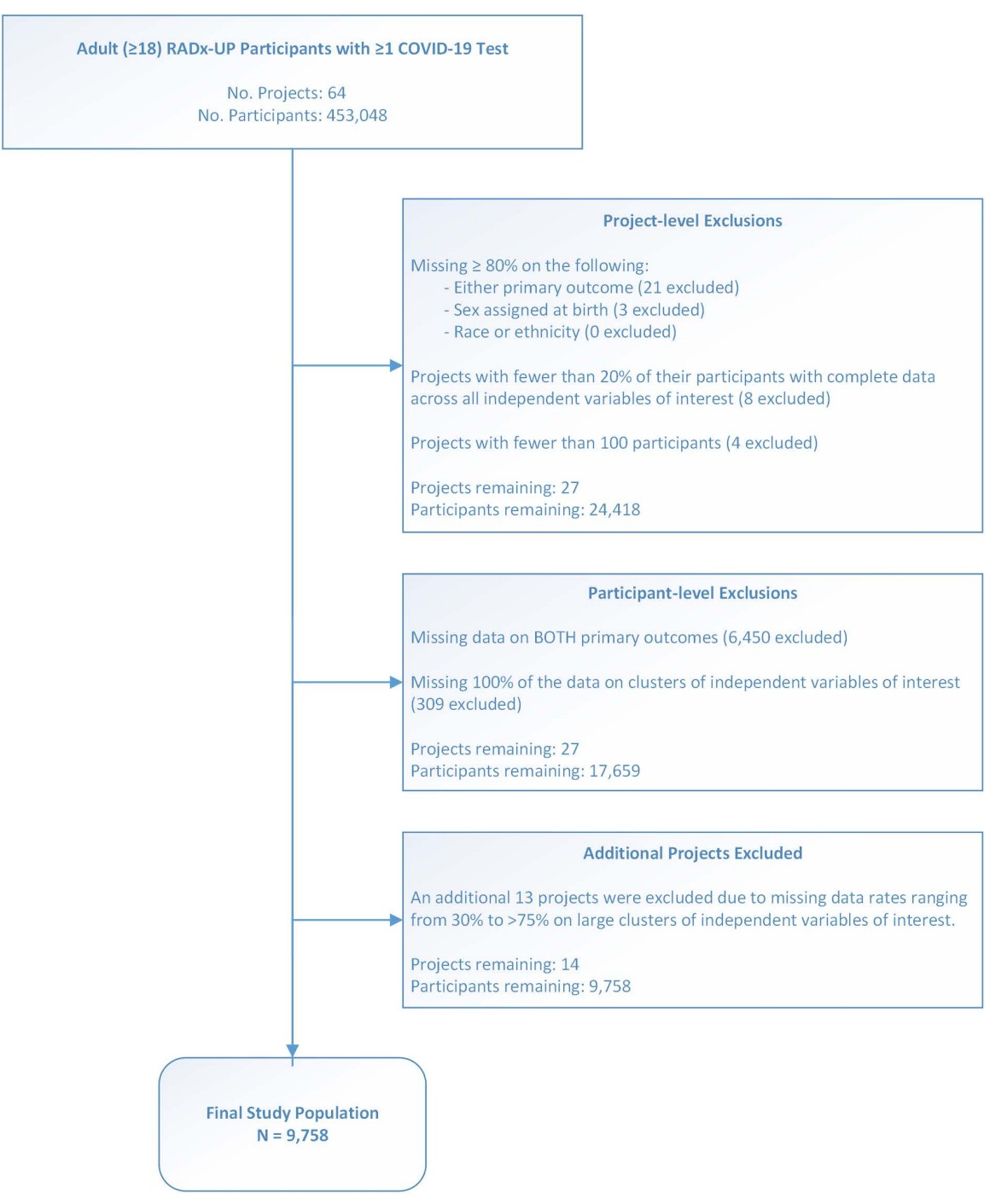

**Fig 1. Derivation of the study population.**

Each of these domains captures interconnected aspects of participants' social environments that may shape their engagement with COVID-19 prevention measures. For example, employment status reflects both economic security and potential exposure settings, while prior flu vaccination history serves as a proxy for healthcare access and established

preventive behaviors. Details about the collection of independent variables are provided in S1 and S2 Table. Description of Prevention Behaviors and Population Characteristics.

## Statistical analysis

Descriptive statistics summarized participant characteristics at the time of first COVID-19 test (baseline). Continuous variables are presented as the number of non-missing values, mean (SD), and median (25th, 75th percentiles). Categorical variables are presented as counts (percentages). The association between independent variables and the primary and secondary outcomes was analyzed using generalized estimating equations with a logit link. Project was included as a clustering group variable in the models to account for geographic and project-based distributions, as RADx-UP projects are specific to certain populations/regions [44]. Among the 14 projects included in the analysis, sample sizes ranged from 95 to 3,200 participants. The percentage of participants who met the primary vaccination outcome and the primary testing outcome ranged from 26.3% to 93.7% across projects and 46.5% to 98.9%, respectively. Within-project correlation was estimated to be 0.04 and 0.02 for the primary vaccine and testing outcomes, respectively, indicating very low correlation among subject measurements from the same project. The assumption of linearity was assessed for all continuous covariates. Age was found to exhibit a nonlinear relationship with the COVID-19 vaccination outcome and, upon visual inspection, was modeled as piecewise linear with a knot point at 55 years. For the COVID-19 testing outcomes, age was modeled assuming a linear relationship with outcome. Multiple imputation with chained equations was performed to impute missing covariate data. Odds ratios (ORs) and 95% confidence intervals (CIs) were reported. Hypothesis testing was two-sided and Wald tests were used at an alpha level of 0.05. Analyses were conducted using SAS v9.4 (Cary, NC).

To evaluate differences in preventive behavior adoption, we used our estimated models to predict the probability of vaccination and testing across representative population groupings. Three risk groups were considered: 1) geo-demographic risk groups, consisting of age, race, and geographic region of residence; 2) economic risk groups, consisting of education level, employment status, and number of economic challenges present (i.e., ability to get access to food, water, transportation, housing, needed medications, health care); 3) health risk groups, consisting of drug use (i.e., intravenous or any substance abuse), cardiovascular disease risk factors (i.e., smoking, vaping, overweight, diabetes, hypertension, and history of cardiovascular disease), and flu vaccination status. For each grouping, we compared populations with higher versus lower likelihood of engaging in these preventive measures (vaccination and testing). These groupings were selected based on commonly occurring combinations of characteristics in our dataset and were chosen to illustrate key differences in preventive measure engagement across demographic, economic, and health-related factors.

## Results

### Study population characteristics

The study population included 9,758 participants from diverse racial, ethnic, and socioeconomic backgrounds (Table 1). The median age was 50 years (IQR: 37–60), and 55.0% were female. The sample was racially and ethnically diverse, with 39.6% Non-Hispanic Black, 34.3% Hispanic, 20.1% Non-Hispanic White, and 6.1% Non-Hispanic Other participants. Socioeconomic disadvantage was prevalent, with two-thirds (66.1%) reporting annual household incomes below $25,000 and 62.7% being unemployed at the time of the study.

### COVID-19 vaccination, testing, and positivity rates

Overall, 72.5% of participants reported receiving a COVID-19 vaccine, and 82.1% (95% CI, 81.3%−82.9%) reported ever having been tested for COVID-19. Among those tested, 26.0% reported a positive result at some point (Table 1). These rates suggest relatively high engagement with COVID-19 prevention measures in the study population.

**Table 1. Summary of population characteristics and prevention behaviors.**

| Characteristic | No. Participants (N = 9,758) |
|---|---|
| **Primary Outcome** | |
| Received COVID-19 Vaccine | 7,079/9,758 (72.5%) |
| **Secondary Outcomes** | |
| Ever Previously Been Tested for COVID-19 | 7,616/9,274 (82.1%) |
| Ever Previously Tested Positive for COVID-19[1] | 1,983/7,616 (26.0%) |
| **Geo-Demographic Characteristics** | |
| Age (yrs.) | |
| N | 9,758 |
| Mean (SD) | 48.3 (14.5) |
| Median (25th, 75th) | 50 (37, 60) |
| Female Sex | 5,322/9,684 (55.0%) |
| Race/Ethnicity | |
| Non-Hispanic, White | 1,929/9,599 (20.1%) |
| Non-Hispanic, Black | 3,797/9,599 (39.6%) |
| Non-Hispanic, Other | 585/9,599 (6.1%) |
| Hispanic | 3,288/9,599 (34.3%) |
| Region | |
| West | 1,792/9,758 (18.4%) |
| Midwest | 3,929/9,758 (40.3%) |
| Northeast | 843/9,758 (8.6%) |
| South | 3,194/9,758 (32.7%) |
| **Economic Characteristics** | |
| Education | |
| Less than High School | 2,370/9,409 (25.2%) |
| High School/GED Completed | 2,678/9,409 (28.5%) |
| Beyond High School | 4,361/9,409 (46.3%) |
| Household Income | |
| <$25,000 | 4,934/7,465 (66.1%) |
| $25,000–$49,000 | 1,257/7,465 (16.8%) |
| $50,000 or More | 1,274/7,465 (17.1%) |
| Current Employment | |
| Not Employed | 5,788/9,229 (62.7%) |
| Employed, Essential Worker | 1,822/9,229 (19.7%) |
| Employed, Non-Essential Worker | 1,619/9,229 (17.5%) |
| Housing | |
| Housed, Lives Alone | 2,959/9,404 (31.5%) |
| Housed, Lives with Others | 4,890/9,404 (52.0%) |
| Unhoused | 1,555/9,404 (16.5%) |
| Economic Challenge, Categories of No. of Challenges | |
| No Challenges | 5,224/9,522 (54.9%) |
| 1 or 2 Challenges | 2,177/9,522 (22.9%) |
| 3 or More Challenges | 2,121/9,522 (22.3%) |
| **Health-Related Characteristics** | |
| Uninsured | 1,133/8,995 (12.6%) |
| Fair-Poor Health | 2,272/9,430 (24.1%) |

*(Continued)*

**Table 1.** (Continued)

| Characteristic | No. Participants (N = 9,758) |
|---|---|
| Disability | 1,944/9,294 (20.9%) |
| Any Drug Use | 1,331/9,453 (14.1%) |
| Any Heavy Alcohol Use | 541/9,113 (5.9%) |
| Any Mental Health Risk | 2,879/9,455 (30.4%) |
| CVD Risk, Categories of Risk Factors | |
| 0, 1, or 2 Risk Factors | 8,110/9,678 (83.8%) |
| 3 or More Risk Factors | 1,568/9,678 (16.2%) |
| Chronic Condition Risk, Categories of Risk Factors | |
| 0 or 1 Risk Factors | 8,515/9,621 (88.5%) |
| 2 or More Risk Factors | 1,106/9,621 (11.5%) |
| Access to COVID-19 Testing | |
| No | 1,140/9,369 (12.2%) |
| Yes | 8,229/9,369 (87.8%) |
| Ever Received Flu Vaccine | 6,730/9,360 (71.9%) |

[1]Among participants who have previously been tested for COVID-19

## Factors associated with vaccination behavior

Multivariable GEE analysis (Table 2) identified several factors significantly associated with vaccination status. *Geographic Context*: Individuals in the Midwest (OR, 2.06; 95% CI, 1.68–2.51; P < .01) and Northeast (OR, 2.62; 95% CI, 1.18–5.84; P < .01) had higher odds of vaccination compared to those in the South. *Race/Ethnicity*: Compared to Non-Hispanic White participants, Non-Hispanic Black participants had lower odds of vaccination (OR, 0.62; 95% CI, 0.51–0.75; P < .01), while Hispanic participants had higher odds (OR, 1.34; 95% CI, 1.11–1.62; P < .01). *Education*: Those with education beyond high school had higher odds of vaccination compared to those with less than high school education (OR, 1.37; 95% CI, 1.17–1.61; P < .01). *Employment*: Essential (OR, 1.40; 95% CI, 1.26–1.55; P < .01) and non-essential workers (OR, 1.37; 95% CI, 1.21–1.54; P < .01) had higher odds of vaccination compared to unemployed individuals. *Economic Challenges*: Participants facing 1 or 2 economic challenges had lower odds of vaccination compared to those without economic challenges (OR, 0.82; 95% CI, 0.70–0.97; P = .05). *Drug Use*: Those reporting drug use had lower odds of vaccination compared to those not reporting drug use (OR, 0.79; 95% CI, 0.67–0.93; P < .01). *Flu Vaccination*: Participants with prior flu vaccination had higher odds of COVID-19 vaccination compared to those who had never received a flu vaccine (OR, 3.54; 95% CI, 2.78–4.51; P < .01). *Cardiovascular Risk*: Participants with 3 or more cardiovascular disease risk factors had higher odds of vaccination compared to those with 0−2 risk factors (OR, 1.32; 95% CI, 1.15–1.52; P < .01).

## Factors associated with testing behavior

Analysis of testing patterns (Table 2) revealed similar social influences. *Geographic Context*: Northeast and Midwest residents had significantly higher odds of being tested than those in the south (OR, 6.62; 95% CI, 3.15–13.93; P < .01 and OR, 2.60; 95% CI, 1.35–5.04; P < .01, respectively). *Sex*: Females had higher odds of being tested compared to males (OR, 1.37; 95% CI, 1.17–1.62; P < .01). *Race/Ethnicity*: Hispanic participants had significantly higher odds of being tested compared to Non-Hispanic White participants (OR, 1.50; 95% CI, 1.17–1.93; P < .01). *Education*: Higher education levels were associated with increased likelihood of testing, with those having education beyond high school having higher odds of testing compared to those with less than high school education (OR, 1.70; 95% CI, 1.39–2.08; P < .01). *Health Insurance*: Uninsured individuals

**Table 2. Social factors associated with COVID-19 prevention behaviors.**

| Characteristic | COVID-19 Vaccination | | | Ever Tested for COVID-19 | | |
|---|---|---|---|---|---|---|
| | Proportion (%) | OR (95% CI) | P-value | Proportion (%) | OR (95% CI) | P-value |
| **Geo-Demographic Risk Characteristics** | | | | | | |
| Age (years), per 5-years[1] | | | | | 0.99 (0.96, 1.02) | 0.47 |
| ≤55 | 4,120/6,162 (66.9%) | 1.15 (1.12, 1.18) | <.01 | --- | | |
| >55 | 2,959/3,596 (82.3%) | 1.27 (1.18, 1.38) | <.01 | --- | | |
| Sex Assigned at Birth | | | | | | |
| Female | 4,005/5,322 (75.3%) | 1.11 (0.97, 1.27) | 0.12 | 4,245/5,322 (79.8%) | 1.37 (1.17, 1.62) | <.01 |
| Male | 3,040/4,362 (69.7%) | Reference | | 3,309/4,362 (75.9%) | Reference | |
| Race/Ethnicity | | | <.01 | | | <.01 |
| Non-Hispanic White | 1,394/1,929 (72.3%) | Reference | | 1,626/1,929 (84.3%) | Reference | |
| Non-Hispanic Black | 2,622/3,797 (69.1%) | 0.62 (0.51, 0.75) | | 2,797/3,797 (73.7%) | 1.15 (0.85, 1.55) | |
| Non-Hispanic Other | 408/585 (69.7%) | 0.97 (0.75, 1.26) | | 454/585 (77.6%) | 0.97 (0.68, 1.39) | |
| Hispanic | 2,551/3,288 (77.6%) | 1.34 (1.11, 1.62) | | 2,619/3,288 (79.7%) | 1.50 (1.17, 1.93) | |
| Region | | | <.01 | | | <.01 |
| West | 1,126/1,792 (62.8%) | 0.77 (0.36, 1.64) | | 1,546/1,792 (86.3%) | 3.41 (0.86, 13.53) | |
| Midwest | 3,141/3,929 (79.9%) | 2.06 (1.68, 2.51) | | 3,376/3,929 (85.9%) | 2.60 (1.35, 5.04) | |
| Northeast | 639/843 (75.8%) | 2.62 (1.18, 5.84) | | 771/843 (91.5%) | 6.62 (3.15, 13.93) | |
| South | 2,173/3,194 (68.0%) | Reference | | 1,923/3,194 (60.2%) | Reference | |
| **Economic Risk Characteristics** | | | | | | |
| Education | | | <.01 | | | <.01 |
| Less than High School | 1,645/2,370 (69.4%) | Reference | | 1,759/2,370 (74.2%) | Reference | |
| High School/GED Completed | 1,788/2,678 (66.8%) | 1.06 (0.93, 1.21) | | 2,043/2,678 (76.3%) | 1.16 (1.04, 1.30) | |
| Beyond High School | 3,450/4,361 (79.1%) | 1.37 (1.17, 1.61) | | 3,660/4,361 (83.9%) | 1.70 (1.39, 2.08) | |
| Household Income | | | 0.14 | | | 0.13 |
| Less than $25,000 | 3,486/4,934 (70.7%) | Reference | | 4,067/4,934 (82.4%) | Reference | |
| $25,000 - $49,999 | 921/1,257 (73.3%) | 0.95 (0.81, 1.12) | | 1,027/1,257 (81.7%) | 1.08 (0.86, 1.35) | |
| $50,000 or More | 1,109/1,274 (87.0%) | 1.17 (0.98, 1.39) | | 1,101/1,274 (86.4%) | 1.29 (1.02, 1.63) | |
| Current Employment | | | <.01 | | | <.01 |
| Not Employed | 4,062/5,788 (70.2%) | Reference | | 4,530/5,788 (78.3%) | Reference | |
| Employed, Essential Worker | 1,419/1,822 (77.9%) | 1.40 (1.26, 1.55) | | 1,480/1,822 (81.2%) | 1.54 (1.31, 1.80) | |
| Employed, Non-Essential Worker | 1,267/1,619 (78.3%) | 1.37 (1.21, 1.54) | | 1,348/1,619 (83.3%) | 1.58 (1.35, 1.84) | |
| Housing | | | 0.52 | | | 0.82 |
| Housed, Lives Alone | 2,133/2,959 (72.1%) | Reference | | 2,307/2,959 (78.0%) | Reference | |
| Housed, Lives with Others | 3,689/4,890 (75.4%) | 1.07 (0.94, 1.23) | | 3,875/4,890 (79.2%) | 0.97 (0.83, 1.12) | |
| Unhoused | 1,024/1,555 (65.9%) | 1.01 (0.85, 1.20) | | 1,249/1,555 (80.3%) | 1.04 (0.92, 1.18) | |
| Economic Challenge, Categories of No. of Challenges | | | 0.05 | | | 0.83 |
| No Challenges | 4,038/5,224 (77.3%) | Reference | | 4,060/5,224 (77.7%) | Reference | |
| 1 or 2 Challenges | 1,559/2,177 (71.6%) | 0.82 (0.70, 0.97) | | 1,794/2,177 (82.4%) | 1.06 (0.88, 1.27) | |
| 3 or More Challenges | 1,323/2,121 (62.4%) | 0.83 (0.68, 1.01) | | 1,686/2,121 (79.5%) | 1.00 (0.84, 1.20) | |
| **Health Risk Characteristics** | | | | | | |
| Uninsured | | | | | | |
| Yes | 749/1,133 (66.1%) | 0.83 (0.61, 1.13) | 0.23 | 822/1,133 (72.6%) | 0.76 (0.64, 0.90) | <.01 |
| No | 5,881/7,862 (74.8%) | Reference | | 6,352/7,862 (80.8%) | Reference | |

*(Continued)*

**Table 2.** (Continued)

| Characteristic | COVID-19 Vaccination | | | Ever Tested for COVID-19 | | |
|---|---|---|---|---|---|---|
| | Proportion (%) | OR (95% CI) | P-value | Proportion (%) | OR (95% CI) | P-value |
| Fair-Poor Health | | | | | | |
| Yes | 1,622/2,272 (71.4%) | 1.03 (0.95, 1.10) | 0.50 | 1,899/2,272 (83.6%) | 1.26 (1.10, 1.45) | <.01 |
| No | 5,268/7,158 (73.6%) | Reference | | 5,598/7,158 (78.2%) | Reference | |
| Disability | | | | | | |
| Yes | 1,382/1,944 (71.1%) | 0.98 (0.87, 1.11) | 0.76 | 1,635/1,944 (84.1%) | 1.10 (0.89, 1.35) | 0.37 |
| No | 5,399/7,350 (73.5%) | Reference | | 5,776/7,350 (78.6%) | Reference | |
| Any Drug Use | | | | | | |
| Yes | 790/1,331 (59.4%) | 0.79 (0.67, 0.93) | <.01 | 1,157/1,331 (86.9%) | 1.19 (1.03, 1.39) | 0.02 |
| No | 6,074/8,122 (74.8%) | Reference | | 6,300/8,122 (77.6%) | Reference | |
| Any Heavy Alcohol Use | | | | | | |
| Yes | 385/541 (71.2%) | 0.92 (0.80, 1.07) | 0.27 | 453/541 (83.7%) | 0.96 (0.82, 1.12) | 0.59 |
| No | 6,262/8,572 (73.1%) | Reference | | 6,819/8,572 (79.5%) | Reference | |
| Any Mental Health Risk | | | | | | |
| Yes | 2,082/2,879 (72.3%) | 1.17 (1.01, 1.35) | 0.04 | 2,505/2,879 (87.0%) | 1.27 (1.11, 1.46) | <.01 |
| No | 4,779/6,576 (72.7%) | Reference | | 4,965/6,576 (75.5%) | Reference | |
| CVD Risk, Categories of Risk Factors | | | | | | |
| 0, 1, or 2 Risk Factors | 5,753/8,110 (70.9%) | Reference | | 6,272/8,110 (77.3%) | Reference | |
| 3 or More Risk Factors | 1,271/1,568 (81.1%) | 1.32 (1.15, 1.52) | <.01 | 1,328/1,568 (84.7%) | 1.24 (1.12, 1.36) | <.01 |
| Chronic Condition Risk, Categories of Risk Factors | | | | | | |
| 0 or 1 Risk Factors | 6,134/8,515 (72.0%) | Reference | | 6,589/8,515 (77.4%) | Reference | |
| 2 or More Risk Factors | 848/1,106 (76.7%) | 1.05 (0.95, 1.15) | 0.33 | 973/1,106 (88.0%) | 1.14 (0.93, 1.39) | 0.20 |
| Access to COVID-10 Testing | | | | | | |
| Yes | 6,037/8,229 (73.4%) | 1.21 (1.07, 1.37) | <.01 | 6,754/8,229 (82.1%) | 1.84 (1.57, 2.16) | <.01 |
| No | 790/1,140 (69.3%) | Reference | | 752/1,140 (66.0%) | Reference | |
| Ever Received Flu Vaccine | | | | | | |
| Yes | 5,526/6,730 (82.1%) | 3.54 (2.78, 4.51) | <.01 | 5,618/6,730 (83.5%) | 1.71 (1.46, 2.01) | <.01 |
| No | 1,313/2,630 (49.9%) | Reference | | 1,781/2,630 (67.7%) | Reference | |

[2]To account for a non-linear relationship with COVID-19 vaccination status, age was modeled as a piecewise linear spline with knot point at 55 years. For COVID-19 testing, age was modeled linearly.

had lower odds of being tested compared to insured individuals (OR, 0.76; 95% CI, 0.64–0.90; P<.01). *Access to Testing*: Participants with access to COVID-19 testing had higher odds of being tested compared to those without access (OR, 1.84; 95% CI, 1.57–2.16; P<.01). *Health Status*: Participants reporting fair or poor health had higher odds of being tested compared to those reporting good to excellent health (OR, 1.26; 95% CI, 1.10–1.45; P<.01). *Mental Health*: Those with any mental health risk had higher odds of being tested compared to those without mental health risks (OR, 1.27; 95% CI, 1.11–1.46; P<.01). *Cardiovascular Risk*: Individuals with 3 or more CVD risk factors had higher odds of being tested compared to those with 0−2 risk factors (OR, 1.24; 95% CI, 1.12–1.36; P<.01).

## Social patterns in test results

Among participants who had been previously tested for COVID-19, several social and environmental factors were associated with positive test results (Table 3): *Geographic Context*: Individuals in the West (OR, 2.21; 95% CI, 1.46–3.35; P<.01) and Midwest (OR, 1.60; 95% CI, 1.02–2.51; P<.01) were more likely to have tested positive than those in the

**Table 3.  Social and environmental factors associated with COVID-19 infection patterns among tested participants.**

| Characteristic | Ever Tested Positive Proportion (%) | OR (95% CI) | P-value |
|---|---|---|---|
| **Geo-Demographic Risk Characteristics** | | | |
| Age, per 5-years | -- | 0.94 (0.91, 0.98) | <.01 |
| Sex Assigned at Birth | | | |
| Female | 1,237/4,245 (29.1%) | 1.22 (1.08, 1.37) | <.01 |
| Male | 725/3,309 (21.9%) | Reference | |
| Race/Ethnicity | | | <.01 |
| Non-Hispanic White | 426/1,626 (26.2%) | Reference | |
| Non-Hispanic Black | 579/2,797 (20.7%) | 0.94 (0.71, 1.25) | |
| Non-Hispanic Other | 122/454 (26.9%) | 0.92 (0.75, 1.13) | |
| Hispanic | 827/2,619 (31.6%) | 1.34 (1.04, 1.72) | |
| Region | | | <.01 |
| West | 613/1,546 (39.7%) | 2.21 (1.46, 3.35) | |
| Midwest | 776/3,376 (23.0%) | 1.60 (1.02, 2.51) | |
| Northeast | 141/771 (18.3%) | 0.90 (0.44, 1.83) | |
| South | 453/1,923 (23.6%) | Reference | |
| **Economic Risk Characteristics** | | | |
| Education | | | 0.16 |
| Less than High School | 398/1,759 (22.6%) | Reference | |
| High School/GED Completed | 544/2,043 (26.6%) | 1.13 (0.98, 1.31) | |
| Beyond High School | 1,007/3,660 (27.5%) | 1.08 (0.98, 1.19) | |
| Household Income | | | 0.05 |
| Less than $25,000 | 949/4,067 (23.3%) | Reference | |
| $25,000 - $49,999 | 335/1,027 (32.6%) | 1.15 (0.94, 1.41) | |
| $50,000 or More | 312/1,101 (28.3%) | 0.93 (0.76, 1.14) | |
| Current Employment | | | <.01 |
| Not Employed | 998/4,530 (22.0%) | Reference | |
| Employed, Essential Worker | 497/1,480 (33.6%) | 1.36 (1.14, 1.61) | |
| Employed, Non-Essential Worker | 418/1,348 (31.0%) | 1.40 (1.17, 1.68) | |
| Housing | | | 0.22 |
| Housed, Lives Alone | 480/2,307 (20.8%) | Reference | |
| Housed, Lives with Others | 1,162/3,875 (30.0%) | 1.21 (0.96, 1.53) | |
| Unhoused | 290/1,249 (23.2%) | 1.06 (0.92, 1.21) | |
| Economic Challenge, Categories of No. of Challenges | | | 0.73 |
| No Challenges | 1,079/4,060 (26.6%) | Reference | |
| 1 or 2 Challenges | 445/1,794 (24.8%) | 0.97 (0.82, 1.15) | |
| 3 or More Challenges | 436/1,686 (25.9%) | 1.04 (0.85, 1.27) | |
| **Health Risk Characteristics** | | | |
| Uninsured | | | |
| Yes | 220/822 (26.8%) | 0.91 (0.80, 1.04) | 0.17 |
| No | 1,658/6,352 (26.1%) | Reference | |
| Fair-Poor Health | | | |
| Yes | 533/1,899 (28.1%) | 1.23 (1.13, 1.35) | <.01 |
| No | 1,420/5,598 (25.4%) | Reference | |
| Disability | | | |
| Yes | 378/1,635 (23.1%) | 1.03 (0.88, 1.20) | 0.72 |
| No | 1,557/5,776 (27.0%) | Reference | |

*(Continued)*

**Table 3.** (Continued)

| Characteristic | Ever Tested Positive Proportion (%) | OR (95% CI) | P-value |
|---|---|---|---|
| Any Drug Use | | | |
| Yes | 242/1,157 (20.9%) | 0.89 (0.78, 1.01) | 0.07 |
| No | 1,705/6,300 (27.1%) | Reference | |
| Any Heavy Alcohol Use | | | |
| Yes | 88/453 (19.4%) | 0.86 (0.70, 1.06) | 0.16 |
| No | 1,813/6,819 (26.6%) | Reference | |
| Any Mental Health Risk | | | |
| Yes | 597/2,505 (23.8%) | 0.98 (0.84, 1.16) | 0.84 |
| No | 1,354/4,965 (27.3%) | Reference | |
| CVD Risk, Categories of Risk Factors | | | |
| 0, 1, or 2 Risk Factors | 1,674/6,272 (26.7%) | Reference | |
| 3 or More Risk Factors | 307/1,328 (23.1%) | 1.11 (0.89, 1.37) | 0.35 |
| Chronic Condition Risk, Categories of Risk Factors | | | |
| 0 or 1 Risk Factors | 1,734/6,589 (26.3%) | Reference | |
| 2 or More Risk Factors | 240/973 (24.7%) | 1.11 (0.96, 1.28) | 0.16 |
| Access to COVID-10 Testing | | | |
| Yes | 1,756/6,754 (26.0%) | 0.99 (0.85, 1.16) | 0.93 |
| No | 204/752 (27.1%) | Reference | |
| Ever Received Flu Vaccine | | | |
| Yes | 1,454/5,618 (25.9%) | 0.99 (0.86, 1.13) | 0.85 |
| No | 465/1,781 (26.1%) | Reference | |

South. *Age*: Younger age was associated with higher odds of testing positive (OR per 5-year increase, 0.94; 95% CI, 0.91–0.98; P<.01); *Sex*: Females had higher odds of testing positive compared to males (OR, 1.22; 95% CI, 1.08–1.37; P<.01); *Race/Ethnicity*: Hispanics had higher odds of testing positive compared to Non-Hispanic Whites (OR, 1.34; 95% CI, 1.04–1.72; P<.01); *Employment*: Both essential workers (OR, 1.36; 95% CI, 1.14–1.61; P<.01) and non-essential workers (OR, 1.40; 95% CI, 1.17–1.68; P<.01) had higher odds of testing positive compared to unemployed individuals; *Health Status*: Participants reporting fair or poor health had higher odds of testing positive (OR, 1.23; 95% CI, 1.13–1.35; P<.01).

**Polysocial risk analysis of preventive behavior adoption**

Table 4 presents the predicted probabilities of COVID-19 vaccination and testing across geo-demographic, economic, and health risk groups, revealing stark disparities in preventive behavior adoption.

**Geo-demographic factors.** Individuals at higher risk for low preventive behavior engagement (age 45, Non-Hispanic Black, residing in the U.S. South) had a predicted probability of vaccination of 0.50 (95% CI, 0.49–0.52) compared to 0.93 (95% CI, 0.90–0.94) for those at lower risk (age 65, Hispanic, residing in the U.S. Northeast), a difference of 43 percentage points. The disparity in testing was smaller but still substantial, with predicted probabilities of 0.69 (95% CI, 0.66–0.72) and 0.96 (95% CI, 0.94–0.96) for the low and high engagement groups, respectively.

**Socio-economic factors.** Groups with lower preventive behavior adoption (less than high school education, unemployed, experiencing 3+economic challenges) had a predicted probability of vaccination of 0.61 (95% CI, 0.59–0.64) compared to 0.82 (95% CI, 0.81–0.83) for groups with higher adoption (beyond high school education, non-essential

**Table 4. Predicted probabilities (95% CI) of prevention engagement across population groups.**

| Risk Group | N (%) of RADx-UP Sample | Predicted Probabilities | |
|---|---|---|---|
| | | COVID-19 Vaccination | COVID-19 Testing |
| **Geo-Demographic Risk Groups** | | | |
| Higher Preventive Behavior Barriers (Age 45, Non-Hispanic/Black, Residing in U.S. South) | 1,649 (17.2%) | 0.50 (0.49, 0.52) | 0.69 (0.66,0.72) |
| Lower Preventive Behavior Barriers (Age 65, Hispanic, Residing in U.S. Northeast) | 417 (4.3%) | 0.93 (0.90, 0.94) | 0.96 (0.94, 0.96) |
| **Economic Risk Groups** | | | |
| Higher Preventive Behavior Barriers (Less than high school, Not employed, 3+ Economic challenges) | 555 (5.7%) | 0.61 (0.59, 0.64) | 0.80 (0.77, 0.82) |
| Lower Preventive Behavior Barriers (Beyond high school, Non-essential worker, No economic challenges) | 715 (7.3%) | 0.82 (0.81, 0.83) | 0.91 (0.90, 0.92) |
| **Health Risk Groups** | | | |
| Higher Preventive Behavior Barriers (Any drug use, 0–2 CVD risk factors, Never received flu vaccine) | 324 (3.3%) | 0.39 (0.36, 0.42) | 0.84 (0.81, 0.87) |
| Lower Preventive Behavior Barriers (No drug use, 3+ CVD risk factors, Received flu vaccine) | 1,013 (10.4%) | 0.87 (0.86, 0.88) | 0.86 (0.84, 0.88) |

Notes: 1. Probabilities are estimated based on regressions reported in Table 2. For predictions, only variables for the risk group change.

2. Higher and Lower Preventive Behavior Barrier groups are defined based on the combination of factors associated with the likelihood of engaging in COVID-19 vaccination and testing, not the risk of poor health outcomes.

worker, no economic challenges), a 21-percentage point difference. For testing, the predicted probabilities were 0.80 (95% CI, 0.77–0.82) for the low adoption group and 0.91 (95% CI, 0.90–0.92) for the high adoption group.

**Health-related factors.** The largest disparities in vaccination were observed across health-related risk factors. Individuals with lower preventive behavior adoption (any drug use, 0−2 CVD risk factors, never received flu vaccine) had a predicted probability of vaccination of only 0.39 (95% CI, 0.36–0.42) compared to 0.87 (95% CI, 0.86–0.88) for groups with higher adoption (no drug use, 3+ CVD risk factors, ever received flu vaccine), a difference of 48 percentage points. However, testing rates were more similar between the high and low health risk groups, with predicted probabilities of 0.86 (95% CI, 0.84–0.88) and 0.84 (95% CI, 0.81–0.87), respectively.

## Discussion

Our findings demonstrate that a polysocial risk framework can capture the complex, compounded effects of geo-demographic, economic, and health-related factors on COVID-19 preventive behaviors with remarkable precision. The data reveal that individuals facing multiple intersecting disadvantages are significantly less likely to be vaccinated, which not only corroborates established theoretical models—such as those related to social capital [29,30]—but also provides actionable insights for public health policy. Specifically, these results advocate for the integration of systematic social risk assessments into clinical practice and community-based interventions, thereby enabling targeted, precision public health strategies.

### Interpretation of key findings

The polysocial risk framework revealed dramatic disparities in prevention behaviors across multiple domains:

**Geo-demographic factors**: Non-Hispanic Black participants aged 45 residing in the South had a 43-percentage point lower predicted vaccination probability (0.50) compared to Hispanic participants aged 65 in the Northeast (0.93). This stark difference highlights how the compound effect of race, age, and region may potentially create barriers to prevention.

**Socio-economic factors**: Groups facing multiple economic challenges (less education, unemployment, financial hardship) showed 21-percentage point lower vaccination rates compared to more economically stable groups. This gap highlights the association between economic precarity and preventive health behaviors.

**Health-related factors**: The largest disparities emerged in health-related domains – individuals with substance use, fewer cardiovascular risk factors, and no flu vaccination history were 48-percentage points less likely to be vaccinated than their counterparts. This suggests a strong association between established patterns of healthcare engagement and COVID-19 prevention behaviors.

Notably, testing disparities were narrower than vaccination disparities across risk groups. This likely reflects the success of widespread testing availability through community sites and at-home options [45] and may also reflect the generally higher rates of testing compared to vaccination during the pandemic.

Our results align with prior research demonstrating the profound impact of social determinants on health behaviors [5,46–48], but they also highlight higher-than-expected preventive engagement among underserved groups [49,50]. This may reflect the success of RADx-UP's community-engaged strategies, emphasizing the importance of trusted, locally tailored interventions.

## Public health and policy implications

The study supports integrating social risk assessments in clinical and public health settings [37,51–55] to proactively address compounded vulnerabilities that drive disparities in preventive health behaviors. Our findings highlight significant disparities in COVID-19 vaccination and testing among different polysocial risk profiles. For example, individuals experiencing intersecting high geo-demographic (Non-Hispanic Black, age 45, Southern residence), economic (low education, unemployment, financial hardship), or health-related risk factors (substance use, low CVD risk, no flu vaccination) were each significantly less likely to be vaccinated compared to groups with low geo-demographic, economic, or health risk profiles.

Healthcare systems can embed standardized polysocial risk screening in clinical encounters, and automated risk stratification tools into electronic health records, enabling targeted outreach to high-risk populations, and creation of risk-based care pathways that trigger specific interventions. The utility of polysocial risk modeling as a predictive tool for identifying populations at highest risk of disengagement from preventive care can inform targeted precision public health interventions.

However, implementation challenges—such as standardization of risk assessment tools and provider training—must be addressed to ensure effective integration into routine care. Findings underscore the need for multi-level interventions that address polysocial risk profiles rather than single risk factors. Beyond COVID-19, the polysocial risk framework has broader applicability for understanding disparities in chronic disease prevention, cancer screening, maternal and child health, and health-related social needs (HRSN) interventions

Healthcare delivery models need to address multiple social needs simultaneously and create sustainable funding mechanisms for social care integration. Community-based initiatives should be prioritized in low vaccine-uptake regions, leveraging trusted local organizations to enhance engagement.

## Strengths, limitations and future research

Polysocial risk scores have limitations, including potential oversimplification, limited generalizability, and lack of direct policy translation [56]. Rather than modeling outcomes with a single poly-social risk score, we examine predicted probabilities of multiple risks. Thus, our study avoids these pitfalls while still providing valuable insights into the cumulative impact of multiple social factors on COVID-19 disparities. Additional strengths of this study include its large, diverse sample of underserved individuals, and examination of multiple COVID-19 outcomes. The use of data from the RADx-UP program provides unique insights into a population often underrepresented in health research.

While this study provides new insights into social risk stratification, several limitations must be acknowledged. Its cross-sectional design limits causal inference, and reliance on self-reported data introduces potential recall and social desirability bias. Data did not include the frequency of COVID-19 testing, which may influence the positivity rates, or the completion of all recommended vaccine doses. The analysis was also constrained to social factors captured through common data elements across RADx-UP studies, which do not fully encompass the breadth of factors affecting underserved communities. While the RADx-UP sample is diverse, it may not entirely reflect all underserved populations, potentially limiting broader applicability. Selection bias is another consideration, as individuals who engage in community-based research may differ from those with less connection to health services, meaning disparities may be even greater than observed. Future research should focus on longitudinal studies to examine how shifting social and economic conditions influence health behaviors over time. Additionally, refining polysocial risk assessment tools—including validation across different populations—will be crucial for scaling its application in clinical and public health settings.

## Conclusions

The polysocial risk framework provides a powerful tool for understanding and addressing disparities in preventive health behaviors. By analyzing COVID-19 vaccination and testing through a multi-dimensional risk model, this study reveals how intersecting social, economic, and behavioral factors drive disparities. Our findings highlight that disparities in prevention are shaped by cumulative social disadvantage, rather than any single factor. Individuals facing multiple overlapping social risks—economic instability, lower healthcare access, and limited preventive health engagement—were significantly less likely to receive vaccines or engage in testing. These patterns emphasize that traditional approaches focusing on singular risk factors miss the broader, systemic forces shaping health behaviors.

Embedding polysocial risk assessment into healthcare and public health systems could fundamentally reshape how we predict, prevent, and intervene in health disparities. By incorporating multi-factorial social risk screening into clinical workflows, public health interventions can become more precise, targeted, and effective—shifting from broad, reactive policies to proactive, data-driven strategies that account for real-world complexity [31,57]. Realizing this potential will require investment in data infrastructure, standardization of risk modeling, and the seamless integration of social risk insights into health interventions.

## Supporting information

**S1 Table. Study project metadata.** This table provides detailed metadata for various study projects, including geographic location, primary and secondary populations, study design, study setting, mode of data collection, and vaccine availability phase. It also includes exclusion criteria for the study population and descriptions of prevention behaviors and population characteristics.
(DOCX)

**S2 Table. Description of prevention behaviors and population characteristics.** This table describes prevention behaviors and population characteristics, including survey questions and derivation details for primary outcomes such as COVID-19 vaccination, testing, and positive test results. It also covers demographics, economic risk characteristics, health risk characteristics, and access to COVID-19 testing.
(DOCX)

**S1 Appendix. Exclusion criteria.** This appendix outlines the exclusion criteria for the study population, including project-level and participant-level exclusions based on missing data and other criteria. It provides detailed information on the reasons for excluding certain projects and participants from the study.
(DOCX)

## Author contributions

**Conceptualization:** David R. Brown, Derek D. Cyr, Lisa Wruck, Troy A. Stefano, Marianna K. Baum, Eileen Fluney, Nana Aisha Garba, Frederick W. Anderson, Haley R. Fonseca, Krista M. Perreira.

**Data curation:** David R. Brown, Derek D. Cyr, Lisa Wruck, Marianna K. Baum, Haley R. Fonseca, Krista M. Perreira.

**Formal analysis:** David R. Brown, Derek D. Cyr, Lisa Wruck, Troy A. Stefano, Nader Mehri, Zoran Bursac, Richard Munoz, Prasad Bhoite, Krista M. Perreira.

**Funding acquisition:** David R. Brown, Marianna K. Baum.

**Investigation:** David R. Brown, Lisa Wruck, Marianna K. Baum, Eileen Fluney, Haley R. Fonseca, Sara Assaf, Krista M. Perreira.

**Methodology:** David R. Brown, Derek D. Cyr, Lisa Wruck, Troy A. Stefano, Nader Mehri, Zoran Bursac, Richard Munoz, Marianna K. Baum, Prasad Bhoite, Nana Aisha Garba, Frederick W. Anderson, Haley R. Fonseca, Krista M. Perreira.

**Project administration:** David R. Brown, Lisa Wruck, Marianna K. Baum, Krista M. Perreira.

**Resources:** David R. Brown, Derek D. Cyr, Lisa Wruck, Nader Mehri, Zoran Bursac, Marianna K. Baum, Eileen Fluney, Krista M. Perreira.

**Supervision:** David R. Brown, Lisa Wruck, Zoran Bursac, Marianna K. Baum, Krista M. Perreira.

**Validation:** David R. Brown, Derek D. Cyr, Lisa Wruck, Nader Mehri, Richard Munoz, Marianna K. Baum, Prasad Bhoite, Krista M. Perreira.

**Visualization:** David R. Brown, Derek D. Cyr, Lisa Wruck, Nader Mehri, Krista M. Perreira.

**Writing – original draft:** David R. Brown, Derek D. Cyr, Krista M. Perreira.

**Writing – review & editing:** David R. Brown, Derek D. Cyr, Lisa Wruck, Troy A. Stefano, Nader Mehri, Zoran Bursac, Richard Munoz, Marianna K. Baum, Eileen Fluney, Prasad Bhoite, Nana Aisha Garba, Frederick W. Anderson, Haley R. Fonseca, Sara Assaf, Krista M. Perreira.

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
