## [Decision Letter · Decision Letter 0]

PONE-D-25-07512COVID-19 Prevention Is Shaped by Polysocial Risk: A Cross-Sectional Study of Vaccination and Testing Disparities in Underserved Populations

PLOS ONE

Dear Dr. Brown,

Thank you for submitting your manuscript to PLOS ONE. After careful consideration, we feel that it has merit but does not fully meet PLOS ONE’s publication criteria as it currently stands. Therefore, we invite you to submit a revised version of the manuscript that addresses the points raised during the review process.

We look forward to receiving your revised manuscript.

Kind regards,

Muhammad Muntazir Mehdi Khan, M.B.B.S.

Academic Editor

PLOS ONE

Journal Requirements:

Research reported in this RADx-UP publication was supported by the National Institutes of Health under Award Numbers U01DA040381, U01MD017423, and U24MD016258. This work was supported by Azure sponsorship credits granted by Microsoft’s AI for Good Research Lab. The content is solely the responsibility of the authors and does not necessarily represent the official views of the National Institutes of Health.

4. In the online submission form, you indicated that this study makes use of data collected by individual RADx-UP projects and shared with the CDCC under the RADx-UP data sharing policy which provides multi-level protections for confidentiality of participants. Data is available by request from the NIH Rapid Acceleration of Diagnostics Data Hub (RADx Data Hub) https://radxdatahub.nih.gov/ .

5. Please remove all personal information, ensure that the data shared are in accordance with participant consent, and re-upload a fully anonymized data set.

Reviewers' comments:

Reviewer's Responses to Questions

**Comments to the Author**

1. Is the manuscript technically sound, and do the data support the conclusions?

Reviewer #1: Yes

Reviewer #2: Yes

2. Has the statistical analysis been performed appropriately and rigorously? 

Reviewer #1: Yes

Reviewer #2: Yes

3. Have the authors made all data underlying the findings in their manuscript fully available?

Reviewer #1: Yes

Reviewer #2: No

4. Is the manuscript presented in an intelligible fashion and written in standard English?

Reviewer #1: Yes

Reviewer #2: Yes

5. Review Comments to the Author

Reviewer #1: Thank you for the opportunity to review this interesting work. Please find my specific feedback below:

Abstract:

• Adequate.

Introduction:

• Very well done! The Introduction clearly identifies the critical gap being addressed by this study. The significance of using the polysocial risk framework is also clearly illustrated by the authors.

Methods:

• There is inconsistency in data collection dates in the Methods section and the abstract.

Methods: Between February 3, 2020 and April 21, 2023.

Abstract: Between October 2020–June 2024.

• “This study used data from the RADx-UP Core Analytic Datasets version 1.6 collected by accessed through the CDCC on April 30, 2023 [40,43]”.

This sentence is grammatically incorrect. The date is again inconsistent; this version says April 30, 2023, while the authors previously noted April 21, 2023, in their Methods.

• Inclusion criteria: The authors note the following inclusion criteria: “Participants were adults aged 18 years and older who underwent COVID-19 testing and were enrolled in a RADx-UP project”

Later, they define one of their outcomes as COVID-19 testing.

Were participants who never tested for COVID-19 included in the study?

• Analysis: Can the authors report the extent of heterogeneity in individual projects included in the study for different outcome analyses? The authors have included Project as a clustering group in their analyses, but it may help if they can report how much heterogeneity was present as well. Just a suggestion though, the analysis in itself is adequate as well.

Results:

• “These characteristics underscore the study's focus on underserved populations disproportionately impacted by the COVID-19 pandemic.”

I believe this sentence would be more appropriate in Discussion rather than Results.

Discussion:

• “This stark difference highlights how race, age, and region compound to create barriers to prevention.”

Please note that this was a cross-sectional analysis of a dataset, which included multiple projects (with very different study designs). Since it did not entirely rely on prospective or randomized designs, it is not appropriate to establish causal inferences. The above sentence implies a cause-and-effect relationship. An alternative way to phrase this would be:

“This stark difference highlights how the compound effect of race, age, and region may potentially create barriers to prevention.”

Alternatively, you can use phrases such as “associated with”, which do not imply causality. Same feedback for further causal statements:

“This gap underscores how economic precarity directly impacts preventive health behaviors.”

“This suggests that established patterns of healthcare engagement strongly predict COVID-19 prevention behaviors.”

• The authors observed the highest disparities in vaccinations when evaluating health-related risk factors, but they also found that testing rates were similar in these groups. The provide the following interpretation:

“This likely reflects the success of widespread testing availability through community sites and at-home options [45].”

However, one may argue that vaccination was also available widespread similar to testing methods. All the other risk groups that the authors studied had consistent disparities in both vaccination and testing practices. It is interesting that the group with highest disparity in vaccination had no differences in testing practices. The authors should provide potential explanations for this.

• Social patterns in test results: Do these patterns reveal individuals with “truly” higher odds of testing positive, or do they simply reflect that these groups were tested more frequently? The patterns of people testing more often and people testing positive are concordant in the authors’ analyses, except for individuals in the Northeast region. These tested more frequently but did not test positive more frequently. What could be a potential explanation for this?

• Limitations:

Self-reported data also introduces concerns related to social desirability bias.

The authors could only study if the participants had ever been tested or not. They could not study the frequency of COVID-19 testing, which may influence the test positivity rates. Similarly, the authors studied whether the participants had ever received a COVID-19 vaccine, they did not study whether participants received all required vaccination doses (the complete series) or not.

Reviewer #2: The authors present a study linking social determinants of health and healthcare disparities to COVID-19 vaccination. Generally the study is well conducted and well reported.

Comments:

- A significant chunk of 'conclusion' can be moved to 'Public Health and Policy Implications' section.

- Authors need to update the Public health and policy implications section in discussion to include more points pertinent to their finding in this study, rather than a general importance of SDOH and Healthcare disparities.

- Details of groupings mentioned in the last paragraph of methods should be described in the methods.

6. PLOS authors have the option to publish the peer review history of their article (what does this mean? ). If published, this will include your full peer review and any attached files.

**Do you want your identity to be public for this peer review?** For information about this choice, including consent withdrawal, please see our Privacy Policy .

Reviewer #1: No

Reviewer #2: No

---

## [Author Response · Author response to Decision Letter 1]

18 Apr 2025

Reviewer #1:

Methods:

• There is inconsistency in data collection dates in the Methods section and the abstract.

o Methods: Between February 3, 2020 and April 21, 2023.

o Abstract: Between October 2020–June 2024.

• “This study used data from the RADx-UP Core Analytic Datasets version 1.6 collected by accessed through the CDCC on April 30, 2023 [40,43]”.

o This sentence is grammatically incorrect. The date is again inconsistent; this version says April 30, 2023, while the authors previously noted April 21, 2023, in their Methods.

Author’s Response: We appreciate the reviewer’s attention to these details. We have revised abstract and methods to clarify that data was collected by individual projects from February 2020 through April 21, 2023, and that the compiled data was accessed through the CDCC on April 30, 2023.

Inclusion criteria:

• The authors note the following inclusion criteria: “Participants were adults aged 18 years and older who underwent COVID-19 testing and were enrolled in a RADx-UP project”

• Later, they define one of their outcomes as COVID-19 testing.

• Were participants who never tested for COVID-19 included in the study?

Author’s Response: We clarified in the measures section that participants were surveyed prior to being tested for the study. In the survey, they were asked whether they had ever been tested for Covid-19. The outcome of never tested reflects their response to the survey prior to being tested for the study.

In the measures section, we clarified that the testing outcome reflected the response to a survey question regarding their previous Covid-19 test prior to the survey. It is not the test result received from the Covid-19 test completed as part of the RADs-UP study.

“COVID-19 testing engagement was assessed through participants' history of testing (“Have you ever been tested for COVID-19?”) prior to their enrollment in the RADx-UP study. Prior test positivity rates were also examined among participants who had previously engaged in testing.”

Analysis:

• Can the authors report the extent of heterogeneity in individual projects included in the study for different outcome analyses? The authors have included Project as a clustering group in their analyses, but it may help if they can report how much heterogeneity was present as well. Just a suggestion though, the analysis in itself is adequate as well.

Author’s Response: Thank you for this suggestion. To this point, we have added this text to the statistical analysis section:

“Among the 14 projects included in the analysis, sample sizes ranged from 95 to 3,200 participants. The percentage of participants who met the primary vaccination outcome and the primary testing outcome ranged from 26.3% to 93.7% across projects and 46.5% to 98.9%, respectively. Within-project correlation was estimated to be 0.04 and 0.02 for the primary vaccine and testing outcomes, respectively, indicating very low correlation among subject measurements from the same project.”

Results:

• “These characteristics underscore the study's focus on underserved populations disproportionately impacted by the COVID-19 pandemic.”

• I believe this sentence would be more appropriate in Discussion rather than Results.

Author’s Response: We removed this statement.

Discussion:

• “This stark difference highlights how race, age, and region compound to create barriers to prevention.”

• Please note that this was a cross-sectional analysis of a dataset, which included multiple projects (with very different study designs). Since it did not entirely rely on prospective or randomized designs, it is not appropriate to establish causal inferences. The above sentence implies a cause-and-effect relationship. An alternative way to phrase this would be: “This stark difference highlights how the compound effect of race, age, and region may potentially create barriers to prevention.”

Alternatively, you can use phrases such as “associated with”, which do not imply causality.

Author’s Response: We appreciate this feedback and have adopted the reviewer’s suggestion.

• Same feedback for further causal statements:

Author’s Response: We appreciate this feedback and made appropriate edits to highlight association rather than causation

“This gap highlights the association between economic precarity and preventive health behaviors.”

“This suggests a strong association between established patterns of healthcare engagement and COVID-19 prevention behaviors.”

• The authors observed the highest disparities in vaccinations when evaluating health-related risk factors, but they also found that testing rates were similar in these groups. The provide the following interpretation:

• “This likely reflects the success of widespread testing availability through community sites and at-home options [45].”

• However, one may argue that vaccination was also available widespread similar to testing methods. All the other risk groups that the authors studied had consistent disparities in both vaccination and testing practices. It is interesting that the group with highest disparity in vaccination had no differences in testing practices. The authors should provide potential explanations for this.

Author’s Response: We realize that our initial statement was a little unclear. We clarified that the testing gap was narrower across all groups than the vaccination gap. While it was most pronounced in the health risk groups, it was also seen in other risk groupings. Our updated language includes:

“Notably, testing disparities were narrower than vaccination disparities across risk groups. This likely reflects the success of widespread testing availability through community sites and at-home options [45] and may also reflect the generally higher rates of testing compared to vaccination in our sample.”

• Social patterns in test results: Do these patterns reveal individuals with “truly” higher odds of testing positive, or do they simply reflect that these groups were tested more frequently? The patterns of people testing more often and people testing positive are concordant in the authors’ analyses, except for individuals in the Northeast region. These tested more frequently but did not test positive more frequently. What could be a potential explanation for this?

Author’s Response: This is a limitation and is now noted in the limitations sections. Of note, the northeast had the highest rates of having been testing and lowest positivity rates. One might suspect they were tested more frequently in the northeast. But we didn’t capture that information.

Limitations:

• Self-reported data also introduces concerns related to social desirability bias.

The authors could only study if the participants had ever been tested or not. They could not study the frequency of COVID-19 testing, which may influence the test positivity rates. Similarly, the authors studied whether the participants had ever received a COVID-19 vaccine, they did not study whether participants received all required vaccination doses (the complete series) or not.

Author’s Response: These points have been added to the updated limitations section

“While this study provides new insights into social risk stratification, several limitations must be acknowledged. Its cross-sectional design limits causal inference, and reliance on self-reported data introduces potential recall and social desirability bias. Data did not include the frequency of COVID-19 testing, which may influence the positivity rates, or the completion of all recommended vaccine doses.”

Reviewer #2:

Conclusion

• A significant chunk of 'conclusion' can be moved to 'Public Health and Policy Implications' section.

Author’s Response: We have moved relevant parts of the conclusion to the Public Health and Policy Implications section and trimmed the conclusion accordingly.

Public Health and Policy Implications

• Authors need to update the Public health and policy implications section in discussion to include more points pertinent to their finding in this study, rather than a general importance of SDOH and Healthcare disparities.

Author’s Response: We have expanded this section to include more specific points related to our findings, such as the significant disparities in COVID-19 vaccination and testing among different polysocial risk profiles.

The following was added to this section:

“Our findings highlight significant disparities in COVID-19 vaccination and testing among different polysocial risk profiles. For example, individuals experiencing intersecting high geo-demographic (Non-Hispanic Black, age 45, Southern residence), economic (low education, unemployment, financial hardship), or health-related risk factors (substance use, low CVD risk, no flu vaccination) were each significantly less likely to be vaccinated compared to groups with low geo-demographic, economic, or health risk profiles.

The utility of polysocial risk modeling as a predictive tool for identifying populations at highest risk of disengagement from preventive care can inform targeted precision public health interventions.

Beyond COVID-19, the polysocial risk framework has broader applicability for understanding disparities in chronic disease prevention, cancer screening, maternal and child health, and health-related social needs (HRSN) interventions.”

Methods

• Details of groupings mentioned in the last paragraph of methods should be described in the methods.

Author’s Response: We have edited this paragraph for clarity.

“To evaluate differences in preventive behavior adoption, we used our estimated models to predict the probability of vaccination and testing across representative population groupings. Three risk groups were considered: 1) geo-demographic risk groups, consisting of age, race, and geographic region of residence; 2) economic risk groups, consisting of education level, employment status, and number of economic challenges present (i.e., ability to get access to food, water, transportation, housing, needed medications, health care); 3) health risk groups, consisting of drug use (i.e., intravenous or any substance abuse), cardiovascular disease risk factors (i.e., smoking, vaping, overweight, diabetes, hypertension, and history of cardiovascular disease), and flu vaccination status. For each grouping, we compared populations with higher versus lower likelihood of engaging in these preventive measures (vaccination and testing). These groupings were selected based on commonly occurring combinations of characteristics in our dataset and were chosen to illustrate key differences in preventive measure engagement across demographic, economic, and health-related factors.”

Editorial guidance:

Figures

Author’s Response: We have completed this step and uploaded the adjusted files. PACE made these adjustments:

• Resolution is changed to 300 PPI

• TIF file is converted to a valid TIF file.

Supporting information

• Please include captions for your Supporting Information files at the end of your manuscript, and update any in-text citations to match accordingly. Please see our Supporting Information guidelines for more information: http://journals.plos.org/plosone/s/supporting-information.

Author’s Response: We have included Supporting Information Captions at the end of the manuscript as follows:

“S1 Supporting Information: This file includes the following:

S1 Table. Study Project Metadata. This table provides detailed metadata for various study projects, including geographic location, primary and secondary populations, study design, study setting, mode of data collection, and vaccine availability p

hase. It also includes exclusion criteria for the study population and descriptions of prevention behaviors and population characteristics

S2 Table. Description of Prevention Behaviors and Population Characteristics. This table describes prevention behaviors and population characteristics, including survey questions and derivation details for primary outcomes such as COVID-19 vaccination, testing, and positive test results. It also covers demographics, economic risk characteristics, health risk characteristics, and access to COVID-19 testing

S1 Appendix. Exclusion Criteria. This appendix outlines the exclusion criteria for the study population, including project-level and participant-level exclusions based on missing data and other criteria. It provides detailed information on the reasons for excluding certain projects and participants from the study”

---

## [Decision Letter · Decision Letter 1]

COVID-19 Prevention Is Shaped by Polysocial Risk: A Cross-Sectional Study of Vaccination and Testing Disparities in Underserved Populations

PONE-D-25-07512R1

Dear Dr. Brown,

We’re pleased to inform you that your manuscript has been judged scientifically suitable for publication and will be formally accepted for publication once it meets all outstanding technical requirements.

Kind regards,

Muhammad Muntazir Mehdi Khan, M.B.B.S.

Academic Editor

PLOS ONE

Additional Editor Comments (optional):

Reviewers' comments:

Reviewer's Responses to Questions

**Comments to the Author**

1. If the authors have adequately addressed your comments raised in a previous round of review and you feel that this manuscript is now acceptable for publication, you may indicate that here to bypass the “Comments to the Author” section, enter your conflict of interest statement in the “Confidential to Editor” section, and submit your "Accept" recommendation.

Reviewer #2: All comments have been addressed

Reviewer #3: All comments have been addressed

2. Is the manuscript technically sound, and do the data support the conclusions?

Reviewer #2: Yes

Reviewer #3: Yes

3. Has the statistical analysis been performed appropriately and rigorously? 

Reviewer #2: Yes

Reviewer #3: Yes

4. Have the authors made all data underlying the findings in their manuscript fully available?

Reviewer #2: Yes

Reviewer #3: Yes

5. Is the manuscript presented in an intelligible fashion and written in standard English?

Reviewer #2: Yes

Reviewer #3: Yes

6. Review Comments to the Author

Reviewer #2: Authors applied polysocial risk framework to identify risk factors associated with COVID-19 prevention behaviors. They have adequately addressed all reviewer comments.

Reviewer #3: Comments by reviewer have been addressed by the author and the draft is according to journal guidelines

7. PLOS authors have the option to publish the peer review history of their article (what does this mean? ). If published, this will include your full peer review and any attached files.

**Do you want your identity to be public for this peer review?** For information about this choice, including consent withdrawal, please see our Privacy Policy .

Reviewer #2: No

Reviewer #3: **Yes: ** Zainab Rustam

---

## [Editor Report · Acceptance letter]

PONE-D-25-07512R1

PLOS ONE

Dear Dr. Brown,

I'm pleased to inform you that your manuscript has been deemed suitable for publication in PLOS ONE. Congratulations! Your manuscript is now being handed over to our production team.

Kind regards,

on behalf of

Dr. Muhammad Muntazir Mehdi Khan

Academic Editor

PLOS ONE